# Adoption and Usage of E-Grocery Shopping: A Context-Specific UTAUT2 Model

Ellen Van Droogenbroeck * and Leo Van Hove

Department of Applied Economics, Vrije Universiteit Brussel, Free University of Brussels, 1050 Brussels, Belgium;
Leo.Van.Hove@vub.be
* Correspondence: Ellen.Van.Droogenbroeck@vub.be

**Abstract:** In order to determine how sustainable online grocery shopping is as a practice, it is crucial to have an in-depth understanding of its drivers. This paper therefore validates the Unified Theory of Acceptance and Use of Technology (UTAUT2) in the context of e-grocery and enriches it with five constructs. We exploit a self-administered survey among 560 customers of two Belgian supermarkets and test the model by means of hierarchical multiple regression analysis. We do so not only for the full sample, but also for users and non-users separately. For the full sample, four of the five proposed context-specific constructs—namely, perceived risk, perceived time pressure, perceived in-store shopping enjoyment, and innovativeness—help better explain the intention to adopt or continue to use e-grocery services. In the subsamples, only perceived time pressure and innovativeness add explanatory power, and this only for non-users. In other words, the additional constructs primarily help discriminate between users and non-users. In addition, while the extended model outperforms the original UTAUT2 model for all three samples, the added value of the extended model does not so much lie in a higher explained variance, but rather in a more correct identification of the drivers of BI.

**Keywords:** e-grocery shopping; adoption; UTAUT2; perceived time pressure; habit

## 1. Introduction

Online retailing is one of the most dynamic and fast-growing industries in the world [1]. In the US, e-commerce sales of physical goods totaled $343.1 billion in 2019 and are expected to reach $476.5 billion in 2024 [2]. The European online retail market shows similar trends. With a predicted annual growth rate of 6.7%, sales are projected to increase from $421.9 million in 2020 to $546.7 million in 2024 [3] (forecasts are adjusted for the expected impact of COVID-19). However, the uptake of online shopping proves to be highly product specific [4,5]. Whereas, worldwide, the share of online sales of consumer electronics and household appliances is expected to grow to, respectively, 39% and 31% by 2023, for food the projection is a mere 3% [6]. Furthermore, in the EU28 only 17% of individuals bought food and groceries via the Internet in 2019, compared to 41% for clothes and sport goods [7]. Overall, the current adoption rates indicate that online grocery shopping has so far been more of an evolution than a revolution [5].

This slow adoption is problematic, since online grocery shopping has the potential to address sustainability and other society-relevant concerns [8,9]. An increased utilization rate of e-grocery services might, for example, aid in reducing emission outputs [10,11] and might enhance access to healthy and fresh products, thus mitigating so-called food deserts [12]. It is therefore of crucial importance to assess consumers' hesitation to adopt online grocery shopping, and to further investigate both the barriers and the drivers, so as to determine how sustainable e-grocery could become as a practice. The upheaval caused by the COVID-19 pandemic has made this even more relevant [13,14].

The essence is that for e-grocery services to be sustainable—both in terms of being profitable for e-grocers and in terms of their impact on the environment—they need to operate at scale. Otherwise, delivery vans drive around half-empty and/or have to navigate

suboptimal routes [15,16]. COVID-19 has forced many people around the world to use an online grocery service for the first time or, for those who had already adopted it, rely on it more than before [17–19]. This raises the question whether COVID-19 will prove to be the trigger that was needed to bring e-grocery demand to a higher plateau. In other words, will there be a lasting impact on consumers' grocery shopping preferences? In particular, will the 'forced adopters' continue to make use of e-grocery services once everything returns to normal?

The recent systematic literature review by Martín et al. [9] shows that the number of articles on e-groceries has increased substantially over the years. Still, a number of gaps remain, including the topic of consumer preferences. Additionally, to date, most studies focus on the US, the UK, India, and China; European countries, including Belgium, are under-researched.

In the e-grocery literature so far, researchers mainly rely on one of the many Information Systems (IS) theories—such as the Theory of Reasoned Action (TRA), the Theory of Planned Behavior (TPB), or the Technology Acceptance Model (TAM). However, these models were originally designed to understand technology acceptance by employees in an organizational context. Consequently, they explain adoption and usage of innovations purely from a technological point of view [20]. According to Childers et al. [21] such a technology-oriented perspective is likely to be fundamentally misguided in a consumer context, both for products with strong hedonic attributes as well as in goal-driven environments—such as online grocery shopping. In a non-work setting, other key factors, such as intrinsic motivations, come to the fore and may even outweigh technological characteristics. In a recent study for the Czech Republic, Klepek and Bauerová [5] show that both hedonic and utilitarian factors are crucial for a good understanding of non-buyers' hesitations towards online grocery shopping. Furthermore, whereas in an organizational set-up employees do not need to consider the cost of the technology, this may well be different for consumers [22].

The present paper therefore uses the Unified Theory of Acceptance and Use of Technology (UTAUT2), the 'meta model' developed by Venkatesh et al. [23], the key advantage being that it explicitly includes factors related to the consumer context—namely hedonic motivation, price value, and habit. In addition, given that e-grocery shopping differs from shopping for other products, we try to enrich the UTAUT2 model with five context-specific constructs, namely perceived risk, perceived time pressure, perceived in-store shopping enjoyment, service quality, and innovativeness.

Our high-level research question is thus whether the UTAUT2 model, extended or not, also works in the context of e-grocery shopping. More specifically, we wanted to find out which of the proposed additional constructs, if any, provide value added. On a more practical level, we also examine whether our dedicated technology adoption model—which was estimated in 'normal', pre-COVID times—can provide (tentative) insights about the post-COVID grocery shopping preferences of those consumers who started using e-grocery services during, and because of, the pandemic.

To test our extended model, we apply hierarchical multiple regression analysis on survey data for 560 customers of two Belgian supermarkets with a click-and-collect service and compare the performance of the extended model with that of the original UTAUT2 model. In doing so, we examine more than just the full sample. Inspired by the novel perspective on innovation adoption research introduced by Reinhardt et al. [24], we also examine, first, the behavioral intention of potential adopters (i.e., current non-users), and second, the intention of users to continue to use the e-grocery service. We explicitly compare the relative importance of constructs and relationships between the two samples (and with the full sample), as it has been demonstrated that potential adopters and users differ when it comes to the determinants of behavioral intention [24–26]. In addition, as the online grocery shopping literature has so far mainly focused on behavioral intention [27], we also examine actual usage.

By doing the above, we contribute to the literature in three ways: (1) We are the first to validate the applicability of UTAUT2 in the online grocery shopping context for a developed country, (2) we develop and test a more comprehensive and context-specific framework, and (3) we identify differences in the determinants of the behavioral intention of users and non-users.

The paper proceeds as follows. The next section discusses the literature and sets out our research model and hypotheses. Section 3 explains the methodology and the data collection process. Section 4 presents the composition of the sample and the results of our regression analyses. In Section 5 we relate our findings to the literature. Finally, Sections 6 and 7 present, respectively, the conclusions and limitations of our research.

## 2. Theoretical Background

In this section, we first provide the necessary theoretical background on the UTAUT2 model. In a second subsection, we then develop our research model and formulate our hypotheses.

### 2.1. The UTAUT2 Model

In order to develop the first version of UTAUT, Venkatesh et al. [28] integrated eight prominent technology acceptance models, including TAM, TRA, and TPB (for a discussion of the evolution of the technology acceptance models up to the first version of UTAUT, we refer to Baptista and Oliveira [29] and Martins et al. [30]). The resulting 'unified' model (see top left corner of Figure 1) suggests that performance expectancy (PE), effort expectancy (EE), social influence (SI), and facilitating conditions (FC) have a direct impact on the behavioral intention (BI) to use information systems in an organizational context. BI then determines actual usage behavior, together with FC; see the top half of Table 1 for definitions.

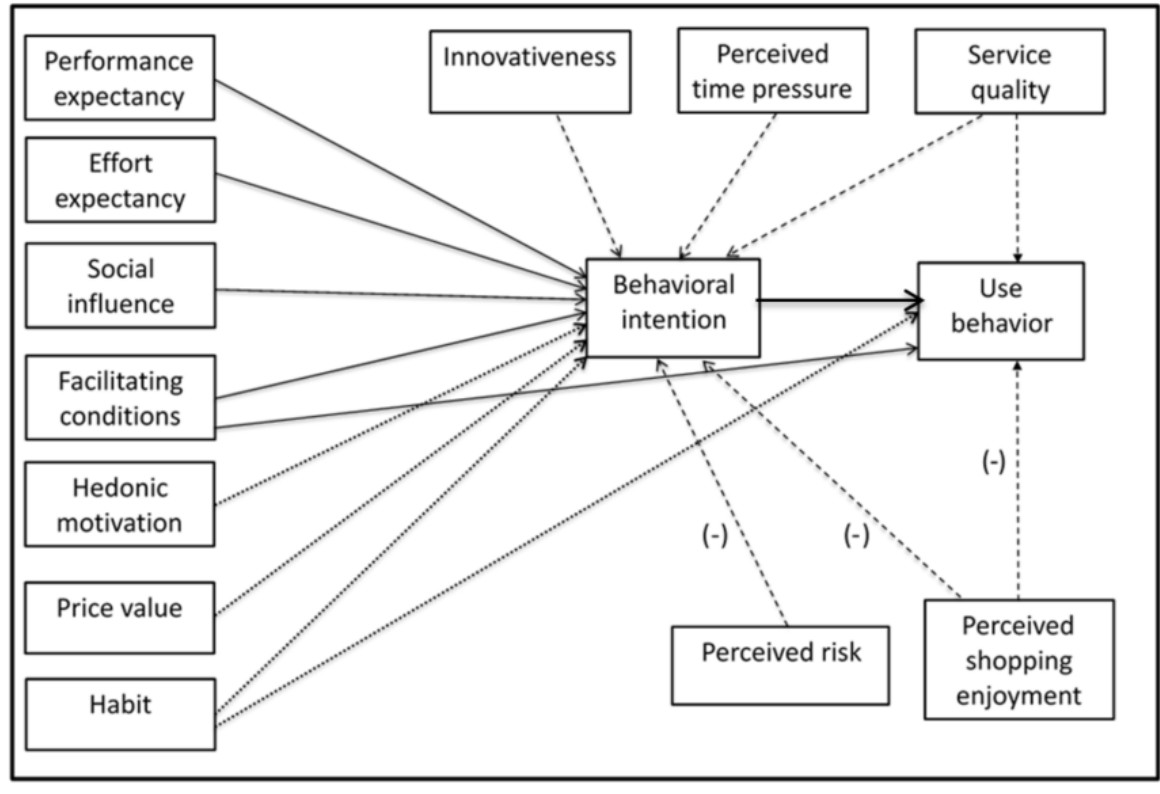

**Figure 1.** Conceptual framework.

**Table 1.** UTAUT and UTAUT2 constructs and their definitions [23].

| Constructs | Definitions |
|---|---|
| *UTAUT* | |
| Performance expectancy | "The degree to which using a technology will provide benefits to consumers in performing certain activities" |
| Effort expectancy | "The degree of ease associated with consumers' use of technology" |
| Social influence | "The extent to which consumers perceive that important others (e.g., family and friends) believe they should use a particular technology" |
| Facilitating conditions | "Consumers' perceptions of the resources and support available to perform a behavior" |
| *Added in UTAUT2* | |
| Hedonic motivation | "The fun or pleasure derived from using a technology" |
| Price value | "Consumers' cognitive tradeoff between the perceived benefits of the applications and the monetary cost for using them" |
| Habit | "The extent to which people tend to perform behaviors automatically because of learning" |

However, like TAM and the other models, UTAUT was originally designed to understand employee technology acceptance in an organizational context. In an attempt to incorporate consumer-context factors, Venkatesh et al. [23] developed a second version of their theory, simply called UTAUT2, by extending it with three constructs (hedonic motivation, price value, and habit) and extra relationships (see the dotted lines in Figure 1).

In their paper, Venkatesh et al. [23] tested both the original UTAUT and their newly developed UTAUT2 for the case of mobile Internet. UTAUT2 was found to explain 74% of the variance in BI and 52% in the variance of usage, while the original UTAUT was able to explain only 56% and 40%, respectively. The higher explanatory power of UTAUT2 in a consumer context is also confirmed in other studies [31], further justifying our choice for UTAUT2 as the starting point for our research model.

UTAUT2 has been applied to several different fields, but in the e-grocery context, it has only been tested in an emerging country setting: Human et al. [32] examine Mauritian consumers' intention to adopt e-grocery services by means of a UTAUT2 model augmented with perceived risk and perceived trust. The present paper is, to the best of our knowledge, the first to use UTAUT2 as the base model for a developed country. Our paper thus answers directly to the call for further research by Venkatesh et al. [23], who recommended to test their model in different countries, age groups, and technologies, and to extend it with other relevant key factors in order to make it applicable to a wide range of consumer contexts. We now discuss our research model in more detail.

### 2.2. Research Model and Hypotheses

As explained in the Introduction, we use an extended version of UTAUT2. Our model includes all the original UTAUT and UTAUT2 constructs; both those explaining behavioral intention (PE, EE, SI, FC, HM, PV, and HB) as well as those explaining actual usage (BI, FC, and HB). In addition, we have included five context variables that would appear relevant in a grocery setting: Perceived risk (PR), perceived time pressure (PTP), perceived in-store shopping enjoyment (PSE), service quality (SQ), and consumer innovativeness (INN); see the dashed lines in Figure 1.

In what follows, we discuss the variables in Figure 1 (and their relationships) in more detail and formulate the corresponding hypotheses. In doing so, we start with the constructs that were already present in the original UTAUT model, but we focus on the constructs that were added in the UTAUT2 round and, in particular, on the constructs that we propose to add.

### 2.2.1. UTAUT Constructs

In their application of UTAUT2 on the case of online grocery shopping in Mauritius, Human et al. [32] find that PE positively influences consumers' BI to start purchasing groceries online, but for EE, SI, and FC, they find no significant impact.

This said, there are e-grocery studies that do not use UAUT but are nevertheless informative, as they examine constructs of other technology adoption models that are similar to UTAUT constructs. Indeed, prior research has already acknowledged the similarities between PE, EE, SI, FC, and constructs from Diffusion of Innovation (DOI), TAM, TRA, and TPB [28,33,34].

Overall, these studies provide substantial evidence for the relationships put forward in the original UTAUT model. For one, papers that include constructs similar to PE (such as perceived usefulness and perceived relative advantage) hint at their potential importance as a predictor of online grocery buying behavior. Concretely, while Hansen [35], Chin and Goh [36], Loketkrawee and Bhatiasevi [37], Nguyen et al. [38], and Frank and Peschel [39] point towards PE's indirect impact via attitude, Verhoef and Langerak [40], Kurnia and Chien [41], Hui and Wan [42], and Driediger and Bhatiasevi [43] find evidence for a direct positive impact on online purchase intentions. Where EE is concerned, several studies explicitly show that the expected effort level and perceived complexity of using online grocery services are significantly negatively related to future buying intentions [35,37–46]. As for SI, the role of the perceived opinion of important reference groups such as friends, colleagues, and family has been confirmed in both qualitative [45,47] and quantitative studies [39,48–52]. Evidence on the impact of FC, however, is less clear-cut. Whereas Hansen et al. [50] and Hansen [49] note that perceived behavioral control (PBC) is positively related to the willingness to use e-grocery services for Swedish consumers, no impact is found for their Danish and German counterparts [50,52]. Finally, the positive relationship between BI and actual usage is well-documented, both in the technology adoption literature [23,28] and in the online shopping context [53–55]. Yet, so far, adoption studies in the online grocery field have used BI mainly as a dependent variable. Boyer and Hult [56] and Kang et al. [57] do examine use behavior, but do not include BI in their model. In fact, only Driediger and Bhatiasevi [43] incorporate BI as a predictor of actual usage and find a significant positive relationship.

Based on the above overview, we proffer the following hypotheses: PE, EE, SI, and FC positively affect the online grocery buying intention (H1, H2, H3, and H4a), and FC and BI positively affect online grocery buying behavior (H4b and H5).

### 2.2.2. UTAUT2 Constructs

- Hedonic motivation (HM):

Psychology research holds that individuals are motivated both extrinsically and intrinsically [58]. Researchers in the technology adoption and online shopping domains have embraced this idea and have examined not only functional (or utilitarian) motivations, such as performance expectancy, but also non-functional (or hedonic) motivations. Venkatesh et al. [23] (p. 161) call it "the fun or pleasure derived from using a technology".

In the online grocery shopping literature, papers that examine perceived enjoyment or HM are rather scarce. Human et al. [32] find evidence for the direct positive influence of HM on Mauritian consumers' BI. In a study for Malaysia, Chin and Goh [36] find a positive impact of perceived enjoyment on consumers' attitude towards online grocery shopping, albeit only at the 10% significance level. In the Thai study of Driediger and Bhatiasevi [43], perceived enjoyment impacts BI via its influence on perceived usefulness and perceived ease of use. Finally, the qualitative research of Ramus and Nielsen [45] shows that HM may hail from consumers' wish to explore a supermarket without the typical stress factors such as crowded aisles, long waiting times at the check-out, and demanding children. Indeed, users of online grocery services state that browsing around and looking for groceries brings them fun and excitement. We thus posit the following hypothesis: HM positively affects online grocery buying intention (H6).

- Price value (PV):

Unlike in an organizational setting, consumers are often obliged to pay fees for services, which can obviously influence their adoption behavior. Studies that examine how consumers trade off collection and delivery attributes in their choice of last mile options find delivery cost to be an important attribute. However, other factors, such as delivery mode, time window, travel time to the grocery store, and return possibilities are found to be (equally) influential [59–61]. These findings relate to the idea in marketing research which holds that consumers compare the costs of goods and services with their quality, in order to determine their 'perceived value' [62] (p. 14). In the UTAUT2 framework, if the perceived benefits (which can vary across consumers) are greater than the monetary costs, PV is positive and adds to the BI to (re-)use the service.

This relationship has been validated in several contexts, such as mobile Internet [23], e-learning [63], m-health [64], and e-commerce [53,55,65]. In the e-grocery literature PV has only been tested once. Human et al. [32], in their study for Mauritius, find evidence for a direct positive impact on potential adopters' purchase intentions. Moreover, in the qualitative study of Van Droogenbroeck and Van Hove [47], several interviewees stated that savings of time and money, together with increased convenience, more than offset the service fee. We therefore posit that PV positively affects online grocery buying intention (H7).

- Habit (HB):

Multiple studies in several contexts, such as e-learning [63], mobile Internet [23], mobile banking [29], mobile TV [66], and online shopping [53,55] have found strong evidence for a significant relationship between HB and BI, and between HB and actual usage. In the e-grocery literature, only Human et al. [32] have already included HB as a predictor variable and find evidence for its positive impact on potential adopters' BI.

Consequently, we hypothesize that HB positively affects online grocery buying intention (H8a), as well as online grocery buying behavior (H8b).

2.2.3. Context-Specific Constructs

We now discuss the five context-specific variables that we propose to add to UTAUT2.

- Perceived Internet Grocery Risk (PR):

Given its importance as a determinant of online behavior [20,54], it is rather surprising that the well-known technology adoption models, including TAM and UTAUT, overlook the perceived risk construct. For online grocery shopping, PR would seem particularly pertinent. Indeed, the fact that consumers cannot see, feel, or taste their groceries themselves, and the resulting concern about the appropriate quality, is an important inhibiting factor for the adoption of online grocery services [8,45,67].

A few researchers in the online grocery literature do include PR in their models. Hansen [35] (p. 39), for example, conceptualizes it as "the extent to which a user believes it is unsafe (e.g., poor security around payment, untrustworthy online shops) to use the web for grocery shopping or that negative consequences (e.g., receiving low quality products, problems with exchanging or returning incorrect items) are possible". Hansen finds that perceived risk is negatively related to both the attitude towards buying groceries online as well as to the online buying intention of Swedish consumers. In another study, for the US, Hansen [48] stresses that the levels of PR do not significantly differ between those who do not use the Internet to shop, those who shop online but not for groceries, and those who use online grocery websites. In a more recent study for Australia, Mortimer et al. [27] confirm that even the most regular online grocery shopper still perceives some level of risk. However, the negative relationship between PR and repurchase intention was only found to be significant for infrequent shoppers. The negative impact of PR is confirmed in studies for Denmark and Sweden [44] and India [68]. Kurnia and Chien [41], Wang and Somogyi [46], Frank and Peschel [39], and Human et al. [32], for, respectively, Australia, China, Denmark, and Mauritius find no impact of PR.

To wrap up, several e-commerce studies have introduced PR into their research, but the construct has not yet been fully examined in the context of online grocery shopping [27]. We therefore add perceived Internet grocery risk—as defined by Hansen [35]—to our model and hypothesize that PR negatively affects online grocery buying intention (H9).

- Perceived time pressure (PTP):

The adage 'too much to do and not enough time' is part and parcel of our contemporary consumer society [69]. In the EU28, only 22.7% of individuals indicate a high satisfaction with their time use [70]. People have the feeling that their discretionary or leisure time has become more restricted and do not like to spend part of this valuable time on obligatory shopping tasks such as grocery shopping [71,72].

In the online shopping literature, PTP is defined as "the degree to which an individual considers him- or herself busy, so that shopping causes high opportunity costs and a feeling emerges of not having enough time to make good purchases" [73] (p. 150). Several authors demonstrate that time pressure affects shopping behavior and preferences [71,74]. Consumers try to perform obligatory shopping tasks as time efficiently as possible, by, for example, visiting stores during low-volume hours, combining shopping tasks [71], visiting 24-hour supermarkets [75], or opting for convenience food [76].

Since one of the major benefits of online grocery shopping is that it saves time [77–79], such services are evidently also a viable option for time-pressured households. To date, only a few—early—studies incorporate perceived time pressure [40,44] or the related 'time available for shopping' [60] in their models. This reasoning leads to the formulation of the following hypothesis: PTP positively affects online grocery buying intention (H10).

- Perceived in-store shopping enjoyment (PSE):

Many consumers perceive shopping for groceries as an unpleasant and stressful task. They approach it from a strictly economic perspective, seeing it as a necessary evil, needed for product acquisition [80,81]. For such consumers, the online solution is particularly attractive. In a recent qualitative study for the US and Germany, the reduction of stressors related to in-store grocery shopping (such as the avoidance of lines, contact with others and crowds, and the need to shop with children) is a commonly cited motive to shop for groceries online [82]. However, other consumers actually enjoy going to physical supermarkets [45]. For these consumers, online grocery shopping will be less appealing or even not at all [40,44,83]. We therefore include, next to the HM of using the online technology, 'perceived in-store shopping enjoyment' as a second intrinsic motivation in our model and define it as "the pleasure one obtains in the shopping process" [84] (p. 174). We posit that PSE negatively affects online grocery buying intention (H11a).

In addition, as most consumers exhibit multi-channel behavior and combine shopping for groceries online with visits to physical supermarkets [45,85,86], PSE might also influence their actual buying behavior. Hence, we posit that PSE negatively affects online grocery buying behavior (H11b).

- Service quality (SQ):

Santos [87] (p. 235) defines e-service quality as "consumers' overall evaluation and judgement of the excellence and quality of e-service offerings in the virtual marketplace". Robinson et al. [88] and Hand et al. [85], in their qualitative studies for the UK, find that issues of SQ are causes of concern for consumers and even possible triggers for discontinuing online grocery shopping. Factors such as late or incomplete orders, unsatisfactory deliveries, bad picking or packing of the goods, and unsuitable replacement items make consumers re-evaluate their decision to shop for groceries online. In a study on click-and-drive services in France, Colla and Lapoule [89] put forward yet another aspect of SQ, namely the friendliness of the staff.

Where quantitative studies are concerned, Boyer and Hult investigate the impact of customers' perceptions of SQ on both BI [90] and actual usage of the online channel [56]. In the first study, for the US, the hypothesis on the positive relationship between SQ and the

BI to re-use the service was strongly supported. In the second study, Boyer and Hult [56] found SQ to be a significant determinant of future buying behavior for an American grocer, but not for a Canadian grocer or for the combined sample. Finally, in a study for Western Europe, Zhu and Semeijn [91] find that SQ positively influences consumers' BI (it is not perfectly clear whether their sample includes only users of online grocery services, or both users and non-users). Based on the above research, we formulate the following hypotheses: SQ positively affects online grocery buying intention (H12a) as well as online grocery buying behavior (H12b).

- Innovativeness (INN):

The literature typically distinguishes between two types of consumer innovativeness, namely general (or global) and domain-specific [92]. Joseph and Vyas [93] (p. 160) conceptualize general innovativeness as "a cognitive style which includes an individual's intellectual, perceptual, and attitudinal characteristics that influences the ways in which he or she reacts to new products, new sensations, new experiences, and communications about them". However, many researchers highlight that innovativeness should be considered as specific to a domain, and not as a general characteristic of an individual's personality [94,95]. Accordingly, Agarwal and Prasad [94] (p. 206) operationalize personal innovativeness in the domain of information technology (PIIT), defined as "the willingness of an individual to try out any new information technology".

In the present paper, we have opted for the domain-specific construct because it has been proven to be more useful in predicting online shopping adoption behavior than general innovativeness [92,95,96]. Several authors have validated the positive effect of domain-specific innovativeness on BI in different contexts. This was done for samples of inexperienced [92,97], experienced [53,98], and a mix of inexperienced and experienced online shoppers [99,100]. Juaneda-Ayensa et al. [101] even find that innovativeness is the strongest predictor of BI.

Yet, to our knowledge, there is no study that has examined the direct impact of (domain-specific) 'innovativeness' on the intention to buy groceries via the Internet. Only Hansen [48] has operationalized consumers' level of general innovativeness by investigating a number of demographics. In light of the current research findings, we hypothesize that INN positively affects online grocery buying intention (H13).

## 3. Materials and Methods

In order to test the hypotheses formulated in the previous section, we collected quantitative data among the customers of two Belgian supermarkets, both users and non-users of e-grocery services. In what follows, Section 3.1 discusses the questionnaire and the measurement scales, Section 3.2 explains the sampling strategy and the data collection, and Section 3.3 introduces the methods for data analysis.

### 3.1. Measurements

The primary data were collected using a self-administered questionnaire in which the constructs of our model were measured through multi-item scales. The items were adapted from the original UTAUT2 model, from existing online grocery research, and from e-commerce adoption studies; see Appendix A. All items were measured using a seven-point Likert scale, ranging from "strongly disagree" (−3) to "strongly agree" (+3). The points on the scale were both numerically and verbally labeled.

As we use our model to explore both intention to adoption (for non-users) and continued use intention (for users), we designed two versions of the questionnaire—in which the wording of the items was modified accordingly. For each group there were also a number of specific questions. For example, in the user version of the questionnaire, actual usage behavior was measured through the self-reported purchase frequency and the average amount spent per order.

Even though the questionnaire was intended for distribution in Dutch and French (the two main languages in Belgium), it was initially drafted in English. It was subsequently

translated into Dutch by a group of researchers and back-translated into English by a bilingual language expert to ensure translation equivalence. The Dutch questionnaire was then translated into French by a bilingual colleague (and back-translated into Dutch by another bilingual colleague). In order to confirm the content validity of the measurement scales, a randomized list of the translated items was given to a group of colleagues, who were asked to assign all items to the matching construct.

Finally, all versions of the survey (Dutch/French, users/non-users, and online/offline) were pilot tested among several subgroups of our population. In total, 17 respondents were asked to provide (oral) feedback. Only small modifications were needed.

### 3.2. Sampling Strategy and Data Collection

The target population of our study was adult customers of the Belgian supermarkets Colruyt and Delhaize for whom the offline and online channels are (roughly) equally attractive in terms of location; that is, who live in a location where their preferred supermarket also has a click-and-collect pick-up point. In Belgium, these pick-up points are usually physically attached to the supermarket, but not all branches have one. Because customer numbers of Belgian supermarkets are not in the public domain and especially because of the additional geographical condition, we cannot provide reliable figures as to the size of our target population. However, in order to give readers a feel for the order of magnitude, let us note that in 2017, 4.2 million people had a loyalty card issued by Colruyt [102]. In the same year, Delhaize had a market share of 23.8%—versus 31.8% for Colruyt Group (Colruyt group includes Colruyt, Okay and Spar)—so for Delhaize, our target population is somewhat smaller [103].

When it comes to sampling, we knew from the outset that it would be impossible to obtain a sampling frame because the Belgian law forbids supermarkets to disclose the identity of their clients. Moreover, relying on such lists would have restricted our sample to customers with a loyalty card. We therefore decided to do the sampling 'on the ground', so to speak. In doing so, we limited ourselves to Brussels and Flanders (the Dutch-speaking part of Belgium). The locations were selected based on two geographical criteria: Region (based on the Nomenclature of Territorial Units for Statistics (NUTS), level 2) and level of urbanization (based on the urban hierarchy of Loopmans et al. [104]). Specifically, we randomly selected one urban and one suburban location in each of the five Flemish provinces. In addition, one (by definition, urban) location was chosen in Brussels. We also made sure that in all locations, the Colruyt and Delhaize supermarkets had a pick-up point.

The questionnaires were distributed at the 11 locations between 17 November and 13 December 2014. The surveys were handed out at the pick-up points and in the stores themselves (on average 8 h per location). Note that, in both cases, the survey instructions specifically asked for the household member primarily responsible for the grocery shopping to complete (the bulk of) the survey. The user version of our questionnaire also contained a short part that had to be filled in by the household member responsible for the picking-up, but these questions are not exploited in the present paper. Respondents could either fill in the questionnaire on paper (and mail it back in a stamped and addressed envelope) or complete the survey online (a link to the online survey was mentioned on the paper copy). In order to ensure the presence of both working and non-working consumers, the data collection was performed on weekdays as well as Saturdays, and at different moments during the day. As an incentive to participate in our survey, there was a raffle, in which 10 respondents could win a €25 voucher for a supermarket of their choice.

In total, we distributed 2382 questionnaires, or on average 213 per location. The total number of questionnaires completed was 638, which amounts to a response rate of 26.8% (the response rate was slightly higher for the non-users than for the users (27.1% versus 25.8%). Interestingly, the majority of the questionnaires (74%) were filled out on paper. After data cleaning, the total number of valid answers available for data analysis was 560: 109 users and 451 non-users.

### 3.3. Data Analysis

In Section 2, we have proposed a comprehensive adoption model for online grocery shopping based on UTAUT2 and five additional variables that are specific for the context. As with other technology acceptance models, our model can be used to analyze the behavioral intention of both potential adopters and current users. However, the origin of the information that consumers use to form beliefs and attitudes, and make decisions, is different for the two groups [24]. In particular, potential adopters have only indirect information about e-grocery shopping while users' intentions and behavior are shaped by first-hand experience with the service. The question is thus whether the drivers of acceptance and continued use are the same.

We therefore test the strength and consistency of our framework not only for the full sample, but—following the approach of Rodríguez del Bosque and Herrero Crespo [105], and Reinhardt et al. [24]—also for two subsamples: One composed of consumers without e-grocery experience, and another formed by users. This approach directly answers the call for further research by other UTAUT researchers [106].

For the full sample and for the potential adopters, we opted for hierarchical multiple regression analysis: We first test a model that includes only the original UTAUT2 variables and subsequently add the context-specific variables PR, PTP, PSE, and INN. For the users, in addition, we employ a system of regression equations. First, the BI to re-use online grocery services is analyzed in the same way as for the other samples (but now with SQ as an additional context-specific variable). The predicted values for BI of the extended model are then used in a regression for actual usage. Because the self-reported order frequency is an ordinal variable, here we use ordered probit. All analyses are performed with the statistical package STATA, version 14.

Prior to the actual estimation of the regression models, we performed multiple preliminary analyses (the full analyses are available upon request). A first remark is that the incidence of missing data in our sample is low. The share per item does not exceed 1.1%, with one exception, namely PSE (with a maximum of 11.1% for PSE1 in the potential adopters sample). Given these low levels of missing data, it was in most cases safe to use the simple mean imputation method to replace them [107]. For PSE, we applied a variant of the two-way imputation method [108]. In particular, we took the average of the person mean and the relevant (user or potential adopter) group mean. Second, the reliability, convergent validity, and discriminant validity of our multi-item scales were assessed using the 'valid scale' command in STATA (see Supplementary Materials). Third, we also carried out the usual tests to check whether the assumptions of regression analysis are valid for our data. The Breusch–Pagan test and plots of the studentized residuals against the fitted values of the dependent variable hinted at the existence of some heteroscedasticity within our dataset (for both potential adopters and users). In what follows, we therefore use White's heteroscedasticity-corrected standard errors—also known as robust standard errors. As this does not change the results of our regressions, heteroscedasticity does not seem to be a serious problem [109].

## 4. Results

This section starts with a descriptive part in which we discuss the composition of our sample. In Section 4.2, we report a first batch of regression results, for the full sample. Subsequently, in Section 4.3, we perform a preliminary analysis in order to test whether potential adopters and users differ on the constructs in our model. Finally, in Section 4.4, we report on the multiple regression analyses for the two subsamples.

### 4.1. Descriptive Analysis

Our sample comprises 560 respondents—109 users and 451 non-users of online grocery services—of which 70.0% are female. The dominance of women is probably the result of our call to have the questionnaire filled out by the household member most often responsible for grocery shopping, combined with the fact that housekeeping is still largely dominated by

women, and that grocery shopping is mainly performed by females [110]. The respondents' age ranges between 18 and 85, with an average of 49 (SD = 14). The median age is 50. The users are significantly younger than the non-users (41.6 versus 50.4 years; t(219) = 7.088, $p \leq 0.001$). The majority of the respondents hold either a professional bachelor (36.1%) or a university degree (24.8%); 46.1% work full-time, 21.6% have a part-time job, and 21.6% are retired. The share of full-time working adults is significantly higher among the users (59.7%) than among the non-users (42.8%), while the share of retired participants is higher in the non-users sample (5.5% versus 25.5%); $\chi^2$ (5; $n = 552$; 27.778, $p \leq 0.001$). Where net income is concerned, the majority of the respondents (60.4%) earn between €1000 and €2500 per month. Finally, about half of the respondents reside in an urban area (55.3%); 83.0% are Dutch speaking. Almost all participants have access to at least one car (90.4%).

Since we lack detailed data about the characteristics of either supermarket visitors or users of online grocery services in Flanders and Brussels, we cannot check the representativeness of our sample. In order to nevertheless give the reader a feel for our data, we have compared our sample with the population of Flanders and Brussels, or—depending on data availability—Belgium. From this point of view, our respondents deviate strongest from the population on the characteristics 'education' and 'occupation'. Compared to the Belgian population, the educational level of the participants is above average. That is, we have relatively many respondents who hold either a professional bachelor (36.1% versus 14.8%) or a university degree (24.8% versus 14.5%). We also observe an overrepresentation of the employed (67.7% versus 52.7%).

Zooming in on the users in our sample (19.5%), a first observation is that they are clearly overrepresented. At the time of our data collection in 2014, only 9% of the Belgian consumers who bought goods or services via the Internet had already used an online grocery service [7]. Note further that the majority of our users had between 1 and 5 years (48.6%) or between 5 and 10 years of experience with ordering groceries online (21.1%). Furthermore, 14.7% only started less than six months ago. The most popular order frequency is 'once every 14 days' (26.6%), followed by 'less than once a month' (25.7%) and 'once a week' (24.8%). Most respondents (68.8%) are multichannel shoppers who combine online orders with regular visits to physical supermarkets. Only 2.8% claim to have stopped buying groceries offline, while 28.4% only go to a physical store when they have forgotten something.

Finally, an important observation is that the preferred service is Collect & Go, the online service of supermarket chain Colruyt. Almost 90% of the respondents (88.1%) indicate that Collect & Go is (one of) the service(s) they use most often (in terms of number of orders). The behavior of our sample thus mainly relates to Collect & Go shoppers. In other words, our findings are not necessarily valid for all users of online grocery services. The popularity of Collect & Go is not that surprising since Colruyt's click-and-collect service is the Belgian pioneer and market leader.

*4.2. Regression Results for the Full Sample*

In what follows, we briefly describe the results of our hierarchical multiple linear regression model for the full sample (see Table 2). The findings are discussed in more detail (and linked with the literature) in the Discussion section. We follow the same approach for the regression results for the subsamples (presented in Section 4.4).

Overall, our model explains the intention to buy groceries online considerably well, with an $R^2$ of 0.53. Note that of all the variables only FC and EE prove insignificant. The latter is significant in the UTAUT2 model but no longer in the extended model. The main predictors of BI are (in order of importance) PE, HM, HB, PR, and PTP. Interestingly, although the additional explanatory power of our extended UTAUT2 model may come across as marginal ($R^2$ change = 0.035), all proposed context-specific variables do significantly impact respondents' BI (note that SQ was only measured for the users). We come back to this point in the Discussion.

**Table 2.** Results of the hierarchical multiple linear regression, full sample (*n* = 560).

| Model 1 DV: Behavioral Intention | Unstandardized Coefficients | | Standardized Coefficients | t | *p*-Value |
|---|---|---|---|---|---|
| | **B** | **Robust Std. Err.** | **Beta** | | |
| Constant | −3.526 *** | 0.411 | | −8.58 | ≤0.001 |
| Performance expectancy (H1) | 0.527 *** | 0.087 | 0.284 *** | 6.09 | ≤0.001 |
| Effort expectancy (H2) | 0.198 * | 0.086 | 0.096 * | 2.30 | 0.022 |
| Social influence (H3) | 0.255 ** | 0.086 | 0.118 ** | 2.96 | 0.003 |
| Facilitating conditions (H4a) | 0.106 | 0.079 | 0.047 | 1.33 | 0.185 |
| Hedonic motivation (H6) | 0.311 *** | 0.076 | 0.172 *** | 4.11 | ≤0.001 |
| Habit (H8a) | 0.248 *** | 0.068 | 0.176 *** | 3.66 | ≤0.001 |
| F (6, 553) = 147.58; $p \leq 0.001$ | | | | | |
| R$^2$ = 0.492 | | | | | |
| Root MSE = 1.650 | | | | | |
| **Model 2 DV: Behavioral Intention** | | | | | |
| Constant | −1.742 * | 0.688 | | −2.53 | 0.012 |
| Performance expectancy (H1) | 0.444 *** | 0.085 | 0.239 *** | 5.22 | ≤0.001 |
| Effort expectancy (H2) | 0.067 | 0.087 | 0.032 | 0.76 | 0.445 |
| Social influence (H3) | 0.206 * | 0.083 | 0.096 * | 2.49 | 0.013 |
| Facilitating conditions (H4a) | 0.031 | 0.079 | 0.014 | 0.39 | 0.694 |
| Hedonic motivation (H6) | 0.337 *** | 0.073 | 0.186 *** | 4.60 | ≤0.001 |
| Habit (H8a) | 0.189 ** | 0.066 | 0.134 ** | 2.88 | 0.004 |
| Perceived risk (H9) | −0.275 *** | 0.074 | −0.132 *** | −3.73 | ≤0.001 |
| Perceived time pressure (H10) | 0.199 *** | 0.054 | 0.121 *** | 3.69 | ≤0.001 |
| Perceived shopping enjoyment (H11a) | −0.112 * | 0.051 | −0.063 * | −2.19 | 0.029 |
| Innovativeness (H13) | 0.125 * | 0.062 | 0.068* | 2.03 | 0.043 |
| F (10, 549) = 103.98; ≤0.001 | | | | | |
| R$^2$ = 0.527 | | | | | |
| Root MSE = 1.598 | | | | | |
| R$^2$ change = 0.035 | | | | | |
| F (4, 549) = 10.060; $p \leq 0.001$ | | | | | |

Note: The constructs PV (H7) and SQ (H12) were only measured for the users, * *p* < 0.050, ** *p* < 0.010, *** *p* ≤ 0.001.

### 4.3. Users Versus Non-Users: Preliminary Analyses

Before we present our regression results for the users and non-users subsamples, in this section, we first perform independent sample t-tests to find out to what extent potential adopters and users differ on the constructs of our model. (Not all groups passed the normality check. Non-parametric Mann–Whitney U-tests were performed as a robustness check. These tests yielded similar results. In addition, whenever the equal variance assumption was violated, we performed unequal variance t-tests.) To that end, Table 3 presents the means and standard deviations of the respective scores. The Likert scores, which range from −3 (strongly disagree) to +3 (strongly agree), were recalculated to a 1–7 scale.

A first, overall observation is that users and potential adopters differ significantly on all constructs and that users have more positive views towards online grocery shopping. Unsurprisingly, the largest difference (4.13 on a 7-point scale) is observed for BI. Users are strongly convinced that they will continue ordering online (mean = 6.55). Conversely, with a mean of 2.42, the BI of potential adopters to start buying groceries online in the near future (within the next year) is definitely on the low side.

Other large discrepancies (i.e., differences > 1) are observed, in descending order, for PE, HB, HM, SI, and PR. For users, ordering online has become a habit (mean = 5.27), while potential adopters do not see this as a possibility for the future (mean = 3.53). The lower level of SI for adopters—that is, the extent to which important others believe the respondent should order groceries online—suggests that the number of their peers who

order online is rather low. The difference in the scores for PR can be explained by the fact that users have already experienced the service and have been able to ascertain that the negative consequences are limited (or non-existent). Still, the PR score for the potential adopter group is neutral (mean = 3.99), which suggests that they do not perceive e-grocery shopping as a particularly risky activity, or that they are simply not aware of the risks involved. At the same time, their PE score indicates that they are also fairly indifferent about the potential benefits, which is in line with their low BI. As for HM, the average score of the users is only 4.46, which indicates that even among users online grocery shopping is not perceived as a particularly pleasant activity.

**Table 3.** Results of the independent sample t-tests.

| Construct | Potential Adopters | | Users | | Difference in Mean |
| | Mean | SD | Mean | SD | (Potential Adopters–Users) |
| --- | --- | --- | --- | --- | --- |
| **UTAUT2** | | | | | |
| PE | 3.83 | 1.03 | 5.58 | 1.00 | −1.75 *** |
| EE | 4.28 | 1.04 | 5.28 | 1.05 | −1.00 *** |
| SI | 3.93 | 0.93 | 5.20 | 1.00 | −1.27 *** |
| FC | 5.44 | 1.04 | 6.13 | 0.75 | −0.69 *** |
| BI | 2.42 | 1.75 | 6.55 | 0.92 | −4.13 *** |
| HM | 3.14 | 1.16 | 4.46 | 1.14 | −1.32 *** |
| PV | / | / | 5.34 | 0.92 | / |
| HB | 3.53 | 1.45 | 5.27 | 1.61 | −1.74 *** |
| **Context** | | | | | |
| PR | 3.99 | 1.03 | 2.95 | 0.98 | 1.04 *** |
| PTP | 4.17 | 1.36 | 5.16 | 1.26 | −0.99 *** |
| PSE | 3.99 | 1.24 | 3.11 | 1.32 | 0.88 *** |
| SQ | / | / | 6.06 | 0.87 | / |
| INN | 3.91 | 1.23 | 4.60 | 1.22 | −0.69 *** |

Note: The constructs PV and SQ were only measured for the users; * $p < 0.050$, ** $p < 0.010$, *** $p \leq 0.001$.

Where the other constructs are concerned, the differences are significant but not that large. For INN, the scores of both potential adopters and users are close to neutral. Given their positive scores for FC, both groups are also confident that they possess the necessary resources to buy groceries via the Internet. The scores for EE show that the degree of ease associated with ordering online is higher among users, but it is not seen as difficult among potential adopters either (mean = 4.28). Finally, users feel more pressed for time than potential adopters, and the level of PSE is rather low for both groups.

*4.4. Users Versus Non-Users: Regression Results*

We now present the results of our regression models for potential adopters and users separately; see Tables 4 and 5, respectively. Overall, the user model explains the intention to continue to purchase groceries via the Internet considerably well, with an $R^2$ of 0.52. For the potential adopters, the $R^2$ is lower (0.29). On the level of the constructs, a first remark is that EE, SI, FC, PV, PR, PSE, and SQ have no direct impact on BI in neither of the two subsamples (EE is significant at the 10% level in the UTAUT2 model but is no longer significant in the extended model).

To focus first on the potential adopters (in Table 4), the main predictors of behavioral intention are (in order of importance): HM, HB (the extent to which non-users believe that online grocery shopping might become an automatic behavior in the future), PTP, and INN (PE is significant in the UTAUT2 model but is no longer significant in the extended model). The first result indicates that BI depends on an intrinsic motivator, i.e., the extent to which potential adopters expect e-grocery buying to be a pleasant activity. For the users (in Table 5), the main predictor of BI is HB. PE is also significant, but only at the 10% level.

To wrap up this section, we also examine users' actual usage of e-grocery services. As mentioned in Section 1, a shortcoming in the literature is that most studies focus on

BI, whereas for e-grocers, it is important to have knowledge not only about customers' re-use intention, but also about which factors determine whether a customer orders online sporadically or on a regular basis. In our survey, use was measured by the self-reported frequency of online ordering, in four categories: Less than once a month, once a month, once every 14 days, or once a week or more. Table 6 shows that the main determinant of use is BI. HB is also significant, but only at the 10% level.

**Table 4.** Results of the hierarchical multiple linear regression, potential adopters (*n* = 451).

| Model 3 DV: Behavioral Intention | Unstandardized Coefficients | | Standardized Coefficients | t | *p*-Value |
|---|---|---|---|---|---|
| | B | Robust Std. Err. | Beta | | |
| Constant | −1.274 ** | 0.429 | | −2.97 | 0.003 |
| Performance expectancy (H1) | 0.148 * | 0.074 | 0.101 * | 2.01 | 0.045 |
| Effort expectancy (H2) | 0.127 ° | 0.075 | 0.086° | 1.68 | 0.093 |
| Social influence (H3) | 0.057 | 0.087 | 0.030 | 0.66 | 0.510 |
| Facilitating conditions (H4a) | 0.055 | 0.076 | 0.033 | 0.71 | 0.475 |
| Hedonic motivation (H6) | 0.336 *** | 0.078 | 0.224 *** | 4.32 | ≤0.001 |
| Habit (H8a) | 0.280 *** | 0.068 | 0.232 *** | 4.14 | ≤0.001 |
| F (6, 444) = 26.91; *p* ≤ 0.001 | | | | | |
| R² = 0.265 | | | | | |
| Root MSE = 1.508 | | | | | |
| **Model 4 DV: Behavioral Intention** | | | | | |
| Constant | −1.357 * | 0.682 | | −1.99 | 0.047 |
| Performance expectancy (H1) | 0.111 | 0.074 | 0.076 | 1.51 | 0.133 |
| Effort expectancy (H2) | 0.087 | 0.074 | 0.059 | 1.17 | 0.243 |
| Social influence (H3) | 0.054 | 0.086 | 0.029 | 0.63 | 0.529 |
| Facilitating conditions (H4a) | 0.008 | 0.078 | 0.005 | 0.10 | 0.918 |
| Hedonic motivation (H6) | 0.324 *** | 0.078 | 0.216 *** | 4.17 | ≤0.001 |
| Habit (H8a) | 0.246 *** | 0.067 | 0.204 *** | 3.68 | ≤0.001 |
| Perceived risk (H9) | −0.079 | 0.074 | −0.047 | −1.07 | 0.287 |
| Perceived time pressure (H10) | 0.151 ** | 0.054 | 0.117 ** | 2.77 | 0.006 |
| Perceived shopping enjoyment (H11a) | −0.001 | 0.052 | −0.000 | −0.01 | 0.990 |
| Innovativeness (H13) | 0.131 * | 0.064 | 0.092 * | 2.04 | 0.042 |
| F (10, 440) = 17.49; ≤0.001 | | | | | |
| R² = 0.287 | | | | | |
| Root MSE = 1.492 | | | | | |
| R² change = 0.022 | | | | | |
| F (4, 440) = 3.423; *p* = 0.009 | | | | | |

Note: ° *p* < 0.100, * *p* < 0.050, ** *p* < 0.010, *** *p* ≤ 0.001.

**Table 5.** Results of the hierarchical multiple linear regression, users (*n* = 109).

| Model 5 DV: Behavioral Intention | Unstandardized Coefficients | | Standardized Coefficients | t | *p*-Value |
|---|---|---|---|---|---|
| | B | Robust Std. Err. | Beta | | |
| Constant | 2.612 ** | 0.952 | | 2.74 | 0.007 |
| Performance expectancy (H1) | 0.333 ** | 0.120 | 0.362 ** | 2.77 | 0.007 |
| Effort expectancy (H2) | 0.099 ° | 0.058 | 0.113 ° | 1.72 | 0.089 |
| Social influence (H3) | 0.063 | 0.074 | 0.069 | 0.85 | 0.396 |
| Facilitating conditions (H4a) | 0.173 | 0.115 | 0.142 | 1.50 | 0.136 |
| Hedonic motivation (H6) | −0.115 | 0.073 | −0.143 | −1.57 | 0.121 |
| Price value (H7) | −0.076 | 0.103 | −0.076 | −0.74 | 0.463 |

**Table 5.** *Cont.*

| Model 5 DV: Behavioral Intention | Unstandardized Coefficients | | Standardized Coefficients | t | *p*-Value |
|---|---|---|---|---|---|
| | **B** | **Robust Std. Err.** | **Beta** | | |
| Habit (H8a) | 0.206 ** | 0.065 | 0.362 ** | 3.16 | 0.002 |
| F (7, 101) = 8.52; *p* ≤ 0.001 | | | | | |
| R² = 0.493 | | | | | |
| Root MSE = 0.676 | | | | | |
| **Model 6 DV: Behavioral intention** | | | | | |
| Constant | 2.612 * | 1.294 | | 2.02 | 0.046 |
| Performance expectancy (H1) | 0.237 ° | 0.126 | 0.257 ° | 1.88 | 0.063 |
| Effort expectancy (H2) | 0.056 | 0.073 | 0.065 | 0.77 | 0.442 |
| Social influence (H3) | 0.041 | 0.072 | 0.045 | 0.57 | 0.573 |
| Facilitating conditions (H4a) | 0.133 | 0.114 | 0.109 | 1.17 | 0.247 |
| Hedonic motivation (H6) | −0.047 | 0.080 | −0.058 | −0.59 | 0.559 |
| Price value (H7) | −0.089 | 0.119 | −0.089 | −0.74 | 0.460 |
| Habit (H8a) | 0.185 ** | 0.068 | 0.324 ** | 2.70 | 0.008 |
| Perceived risk (H9) | −0.040 | 0.082 | −0.050 | −0.48 | 0.630 |
| Perceived time pressure (H10) | 0.092 ° | 0.055 | 0.126 ° | 1.67 | 0.099 |
| Perceived shopping enjoyment (H11a) | −0.039 | 0.051 | −0.056 | −0.77 | 0.442 |
| Service quality (H12a) | 0.149 | 0.137 | 0.141 | 1.08 | 0.281 |
| Innovativeness (H13) | −0.026 | 0.058 | −0.034 | −0.44 | 0.658 |
| F (12, 96) = 5.56; ≤0.001 | | | | | |
| R² = 0.522 | | | | | |
| Root MSE = 0.673 | | | | | |
| R² change = 0.029 | | | | | |
| F (5, 96) = 1.170; *p* = 0.329 | | | | | |

Note: ° *p* < 0.100, * *p* < 0.050, ** *p* < 0.010, *** *p* ≤ 0.001.

**Table 6.** Results of the ordered probit regression analysis, users (*n* = 109).

| | Model 7 DV: Use (Order Frequency) | | Model 8 DV: Use (Order Frequency) | |
|---|---|---|---|---|
| | **Coeff.** | **Robust Std. Err.** | **Coeff.** | **Robust Std. Err.** |
| Facilitating conditions (H4b) | −0.170 | 0.166 | −0.159 | 0.169 |
| Behavioral intention (H5) | 1.210 ** | 0.438 | 1.163 * | 0.558 |
| Habit (H8b) | 0.282 ° | 0.157 | 0.287 ° | 0.173 |
| Perceived shopping enjoyment (H11b) | | | −0.076 | 0.096 |
| Service quality (H12b) | | | −0.027 | 0.177 |
| Pseudo R²: | 0.256 | | Pseudo R²: 0.258 | |
| Log pseudolikelihood: | −112.093 | | Log pseudolikelihood −111.723 | |
| Wald Chi²: | 74.13 *** | | Wald Chi²: 74.81 *** | |

Note: ° *p* < 0.100, * *p* < 0.050, ** *p* < 0.010, *** *p* ≤ 0.001.

## 5. Discussion

The primary objectives of our study were (1) to validate UTAUT2 in the context of online grocery shopping, and (2) to try to improve the model by extending it with additional constructs. Where the first objective is concerned, (the extended version of) UTAUT2 is capable of explaining a considerable portion of the variance in Belgian consumers' BI to order groceries online (R² = 0.53). The same is true for users' BI to re-purchase groceries via the Internet (R² = 0.52).

The BI of potential adopters proves to be less easy to explain, with the $R^2$ ranging between 0.27 and 0.29 (see Table 4). According to the technology adoption literature, adoption is driven by a cognitive elaboration process in which consumers would, prior to adoption, consciously evaluate multiple innovation characteristics [23,111]. This would then result in a certain level of intention to adopt. However, in spite of the evidence that these characteristics—as measured by our constructs—indeed influence the adoption process, it would appear difficult to explain the specific moment of transition from non-adoption to adoption [24]. Indeed, the relatively low $R^2$ in our potential adopters sample implies that a large proportion of the variance (>70%) is not explained by technology adoption factors. In other words, for a high number of non-users, the positive (negative) upfront evaluation of online grocery shopping does not result in a high (low) BI. To actually move from the non-adopter to the adopter state, consumers apparently need a trigger; the presence of adoption drivers and/or the absence of barriers would not seem sufficient [24].

In this respect, Van Droogenbroeck and Van Hove [47], in their qualitative research, find that the first use of an online grocery service is often induced by a situational factor of sorts (such as health problems or the receipt of vouchers for the service cost (the service cost is the fee a customer has to pay every time he or she orders groceries online). These situational factors would seem to be the missing link between the (positive) evaluation and the decision to actually start using the service. For many consumers, it is only once they are confronted with a new situation that they start (or re-start) thinking of the service in terms of the technology adoption constructs. This then leads, as can be inferred from the higher $R^2$ in the users sample, to a better 'pass-through' between construct evaluation and BI.

Turning to our second objective, the extended model explains only a slightly higher proportion of the variance in BI than the original UTAUT2. For actual usage (in Table 6), the $R^2$ is the same for the original UTAUT2 and the extended version because the two context variables—PSE and SQ—are not significant. As can be seen in Table 2, for the full sample the increase in $R^2$ is significant but amounts to only 0.035. A similar remark can be made for the potential adopters, with a significant increase of 0.022. For the users sample, the increase is not significant (see Table 5). At first sight, the added value of an extended UTAUT2 thus seems limited. However, several context variables do prove to be significant—at least for the full sample and for the potential adopters. In fact, in the full sample, all context-specific constructs are found to be predictors of BI. Adding these variables to the model results in either the insignificance of relationships between UTAUT2 constructs and BI (EE → BI) or a decrease in the coefficients and/or significance levels (all other significant UTAUT2 constructs except HM). Looking at the potential adopters sample, it becomes even more clear that the added value of the extended model does not so much lie in a higher explained variance, but rather in a more correct identification of the drivers of BI. For example, as can be seen in Table 4, PE positively impacts the BI of potential adopters in the original UTAUT2 but is no longer significant in the extended model, where the influence of PE would seem to be replaced by a significant relationship between PTP and BI. Similarly, adding INN as the only context-specific variable to the original UTAUT2 model removes the (marginal) significance of EE. This suggests that the relative advantages of e-grocery services as perceived by potential adopters mainly relate to time savings. The same is true for the users, but to a lesser extent. PE positively impacts the BI of users in the original UTAUT2 but is only significant at the 10% level in the extended model, where the influence of PE would seem to be replaced by a significant—also at the 10% level—relationship between PTP and BI. Indeed, when we add PTP as the only context-specific variable in our hierarchical regression model, we observe that PE turns insignificant. In addition, 'low PTP' non-users score significantly lower on PE than those who are 'high PTP' (3.6 versus 4.1; t(449) = -4.787, $p \leq 0.001$). In addition, if we test the original UTAUT2 separately for the 'low PTP' potential adopters, the relationship between PE and BI is not significant (β = 0.018, $p$ = 0.821), whereas for the 'high PTP' group PE is significant (β = 0.185, $p$ = 0.008).

A third contribution of our paper is that we not only test our proposed model for the full sample, but that we also investigate whether the determinants of BI differ between potential adopters and users. We effectively find that the BI of potential adopters is determined by a different (and richer) set of constructs than for the users, which is in line with the findings of Brown et al. [25], but clashes with the results of Rodríguez del Bosque and Herrero Crespo [105], who find that the determinants of BI are the same. In particular, we find that whereas for the potential adopters the BI to start buying groceries online is determined by hedonic motivation, habit, perceived time pressure, and innovativeness, users' BI for continued use is driven by habit and—to a lesser extent—performance expectancy. In more general terms, whereas for users BI is solely determined by utilitarian variables, for non-users hedonic motivations are important as well. This finding clashes somewhat with Reinhardt et al. [24], who find that performance improvements are more frequently cited as crucial adoption triggers by non-adopters.

On the level of the individual constructs, FC, PV, and SQ are the only variables that have no impact in any sample (see Table 7), which leads to a clear rejection of H4a, H7, and H12 (note that PV (H8) and SQ (H12) were only tested for the users). In the full sample, all other proposed relationships are significant, which implies that both the UTAUT2 constructs and the context-specific variables are needed to explain the BI to adopt, c.q. continue to use e-grocery services. In addition, several constructs are only significant in the full sample (SI, PR, PSE, and—to a lesser extent—EE). In other words, these variables discriminate between respondents with a high and a low BI, and thus between potential adopters and users, but do not explain the variance in BI for the respective subsamples (which are more homogenous; see below).

**Table 7.** Overview of the results for the dependent variable behavioral intention (BI).

| DV: BI | Full Sample | | Potential Adopters | | Users | |
|---|---|---|---|---|---|---|
| | UTAUT2 | Extended | UTAUT2 | Extended | UTAUT2 | Extended |
| H1: PE | + *** ($\leq$0.001) | + *** ($\leq$0.001) | + * (0.045) | ns | + ** (0.007) | + ° (0.063) |
| H2: EE | + * (0.022) | ns | + ° (0.093) | ns | + ° (0.089) | ns |
| H3: SI | + ** (0.003) | + * (0.013) | ns | ns | ns | ns |
| H4a: FC | ns | ns | ns | ns | ns | ns |
| H6: HM | + *** ($\leq$0.001) | + *** ($\leq$0.001) | + *** ($\leq$0.001) | + *** ($\leq$0.001) | ns | ns |
| H7: PV | / | / | / | / | ns | ns |
| H8: HB | + *** ($\leq$0.001) | + ** (0.004) | + *** ($\leq$0.001) | + *** ($\leq$0.001) | + ** (0.002) | + ** (0.008) |
| H9: PR | | – *** ($\leq$0.001) | | ns | | ns |
| H10: PTP | | + *** ($\leq$0.001) | | + ** (0.006) | | + ° (0.099) |
| H11a: PSE | | – * (0.029) | | ns | | ns |
| H12: SQ | | / | | / | | ns |
| H13: INN | | + * (0.043) | | + * (0.042) | | ns |

Note: *p*-values in parentheses; ° *p* < 0.100, * *p* < 0.050, ** *p* < 0.010, *** *p* $\leq$ 0.001.

Not surprisingly, differences in BI among either users or non-users prove more difficult to explain. For the latter, we find that HM is the most important determinant. Consumers who expect e-grocery buying to be a pleasant or enjoyable activity are more likely to form intentions to use the online channel. Intrinsic motivations are thus not only important in hedonic settings but are also relevant for utilitarian technologies [112,113]. In addition, potential adopters' BI is also positively influenced by their level of innovativeness. The significance of this relationship is well-documented in the broader technology adoption and e-commerce literature but has—to the best of our knowledge—not yet been validated in the context of online grocery shopping.

HB is the only variable that is significant across all samples: It is the most important determinant of BI for users, the second most important for potential adopters, and the third most important for all respondents. This is not surprising as grocery shopping is a routinized and habitual task [114,115]. The relationship between HB and BI has also been found to be significant in an emerging country context, at least for potential adopters [32].

The insignificance of some of the constructs might be explained by the low variation in the data. For example, in the users sample, SQ has a mean of 6.06 and a variance of only 0.75; 71.6% of the users have a score of 6 or 7. A similar remark can be made for the insignificance of PV and the marginal significance of PTP. In the same vein, 77.1% of the users answered 'very likely' on the BI question "How likely is it that over the next year you will shop for groceries via the Internet?" In fact, only 5.5% have a score lower than 5 out of 7 on this question. In other words, what our model explains are not so much differences between 'low BI' and 'high BI' users, but differences in the 'extent of likeliness', or, in other words, what determines whether a respondent is ('somewhat likely',) 'likely', or 'very likely' to repurchase groceries via the Internet. For the potential adopters, the picture is similar, but in the other direction. Furthermore, 45.6% of the potential adopters indicate that it is very unlikely (score of 1) that they will order groceries online over the next year, while only 14.9% have a score of 5 or higher.

A fourth contribution of our study is that we not only investigate consumers' re-use intention, but also their actual use of the online channel. We hereby respond to calls for more research into actual usage behavior, such as order frequency, in the field of online grocery shopping [27,44]. We find that the sole significant determinant of use is behavioral intention, which is in line with the findings of Driediger and Bhatiasevi [43] for Thailand. The context variables PSE and SQ have no influence on order frequency. The insignificance of SQ clashes with Jara et al. [116], who find that relational variables (i.e., relations between customers and employees, service, and marketing policy) are the key factors that create long-term value for customers of click-and-collect systems in France.

Finally, as noted in the Introduction, on a more practical level we also wanted to examine whether our model might not provide insights as to whether the grocery shopping preferences of the consumers who were more or less forced into adoption during the COVID-19 pandemic have undergone a lasting change or not. At first sight, it would seem improbable that a model that was estimated with pre-COVID data could be useful in this respect. However, on closer scrutiny, the fact that we have results not only for the full sample, but also for non-users and users separately—and in particular, the fact that there prove to be differences between the two groups—does create possibilities. This said, our analysis comes with an important caveat: It is unclear to what extent our findings can be generalized (see Section 7, 'Limitations'). What follows should thus be read as applying to Belgium (and more specifically Flanders), and not necessarily to other countries.

To start with the positive news for e-grocers, for the 'forced adopters' the typical 'innovativeness' and 'perceived risk' barriers to adoption have been overcome and, as the results for the users sample show, are unlikely to re-emerge (provided that new adopters have effectively experienced that their fear for receiving incorrect or low-quality goods was unfounded; see below). Similar remarks can be made about performance expectancy: After having used the service a couple of times, they might realize that it effectively saves them time [117]. Where social influence is concerned, given the increased uptake,

post-COVID more consumers might have more friends and acquaintances who think that buying groceries via the Internet is a good idea. What might also prove important is that the pandemic lasted quite long and that many a country went through several waves and associated lockdowns. As a result, many of the new adopters will probably have used the service several times, perhaps enough to turn it into a habit—which, as the results for the users sample show, is a strong predictor of the intention to continue to use the service.

On the other hand, there is anecdotal evidence that e-grocery services, in Belgium and elsewhere, had—at least initially—difficulties coping with the surge in demand [118,119]. The new adopters may thus not have experienced the service in optimal conditions (inconvenient time slots, longer-than-normal waiting times at the pick-up point, etc.), which may have lowered their performance expectancy (or confirmed their low expectations). Furthermore, consumers who in normal circumstances enjoy shopping in a physical supermarket but who nevertheless started using the online channel (if there are any), are liable to switch back once things return to normal. Finally, there is perceived time pressure, which is an important determinant in both the full and the non-users sample. This is in se an objective factor that will not go away (nor increase) by making use of an e-grocery service, as it is determined by people's work situation, the composition of their household, etc. Hence, in principle, the same logic applies as for shopping enjoyment: New adopters with low time pressure may well give up on online grocery shopping post-COVID. Moreover, if part-time working from home becomes the post-pandemic norm, some of those who started to use the service prior to COVID-19—perhaps in part because of time pressure issues—might actually see their time pressure go down, as working from home brings possibilities for time-shifting.

Clearly, the relative magnitudes of the above effects are uncertain [17,120]—and, as often, the future will tell—but we think that, on balance, there are reasons to believe that the 'COVID shock' will help bring e-grocery services to scale. Indeed, studies that questioned respondents about their post-pandemic shopping behavior find that among those who started to buy (more) groceries online, a considerable share intend to continue this practice in the future [121–123]. However, these studies examine intentions; more research on consumers' actual post-COVID shopping patterns is crucial.

## 6. Conclusions

Compared to the extant literature, this paper proposes a more comprehensive framework for explaining the intention to (re)purchase groceries via the Internet, as well as users' actual usage. The model incorporates all UTAUT2 variables along with five additional constructs that a priori seemed particularly relevant in a grocery shopping context. Data were collected among 560 customers of two Belgian supermarkets and analyzed by means of hierarchical multiple regression analysis—not only for the full sample, but also for users and non-users separately.

For both the full sample and the users sample our extended model predicts BI well. Our model explains considerably less of the variance in potential adopters' BI or in users' order frequency. In the full sample, four of the five proposed context-specific constructs—namely perceived risk, perceived time pressure, perceived in-store shopping enjoyment, and innovativeness—are predictors of BI. In the subsamples, only perceived time pressure and innovativeness add explanatory power, and this only for non-users. In other words, the additional constructs primarily help discriminate between users and non-users. Additionally, while the extended model outperforms the original UTAUT2 model for all three samples, the added value of the extended model does not so much lie in a higher explained variance, but rather in a more correct identification of the drivers of BI.

Overall—that is, taking into account both the original and the newly proposed constructs—the main drivers of users' behavioral intention to repurchase groceries via the Internet are habit and performance expectancy. For the potential adopters, the drivers to start buying groceries online in the future are, in order of importance, hedonic motivation, habit, perceived time pressure, and innovativeness.

To turn to the managerial implications, the difference in determinants highlights the need to—in normal times—develop distinct marketing strategies for potential adopters and those who already order groceries online [24,26,105]. In other words, e-grocers could benefit from emphasizing other aspects of the service. If the target is potential adopters, online grocery shopping could be pictured as a pleasant alternative for the routinized and time-consuming trip to the physical supermarket. Once consumers use the service, the communication could focus on the perceived advantages, such as saving time, saving money, and the convenience of ordering outside opening hours.

The importance of habit for BI (across all samples) not only demonstrates that this construct should be incorporated into technology adoption research if relevant [53,124], but also that e-grocers could benefit from emphasizing the habitual behavior of e-grocery buying in their marketing communication strategies. In a commercial, current users could explain how they have replaced their weekly trip to the physical supermarket by regularly ordering online and picking-up the groceries, for example, while driving home from work. E-grocers should also consider to (further) facilitate and encourage the use of shopping lists, as these allow for easy re-ordering of frequently purchased goods such as milk.

Finally, the wave of new adopters brought about by the COVID-19 pandemic has presented e-grocers with a new challenge, namely retaining as many as possible of these 'forced' adopters. With this perspective in mind, turning online ordering into a habit comes to the fore once again. As shown, habit is not only a key predictor of the intention to continue to use an e-grocery service, but it would also seem the most actionable factor in our above analysis of the longevity of the COVID-19 windfall. Hence, features that can encourage habit formation, such as the shopping lists mentioned earlier, would now seem even more essential than before.

However, overall, our analysis gives reasons to believe that at least part of the increase in demand will prove lasting and will, in this way, help improve the sustainability of online grocery shopping. The positive impact will obviously be bigger for home delivery than for the click-and-collect model that we analyze. However, this is not to say that there are no environmental benefits to be reaped with the latter model. Increased use of click-and-collect services might reduce the number of dedicated trips to the supermarket, given that trip-chaining—on the way home from work or after picking up the children from school, etc.—is easier when the only thing one needs to do is pick up an online order [125].

## 7. Limitations

Our study obviously has a number of (potential) limitations. First, the generalizability of our results has its limits. As we restricted ourselves to Brussels and Flanders, our sample is not representative for the Belgian population. Moreover, the employed and the highly educated are overrepresented. In addition, where the users sample is concerned, the large majority are customers of Collect & Go. On a higher level, our data are limited to the specific case of Belgium. As a consequence, our findings are not necessarily valid for other countries, in which, for example, the click-and-collect model is less popular. More research, in other countries and other cultures, is needed to validate the results.

Second, our dataset stems from late 2014. However, we are convinced that this is not much of an issue, at least not for Belgium. Indeed, in Belgium the uptake of e-grocery services did not change dramatically between 2015 and early 2020. The share of individuals who have bought groceries online increased only from 6% to 11% [7]. Moreover, click-and-collect services are still dominant.

Third, we do not take into account the possible impact of socio-demographics as moderating variables. Further research could therefore perform a multi-group analysis and test how the relationships in our model may differ among consumers at different life stages [25,126]. For example, whereas effort expectancy is not significant for the full sample, it might determine the behavioral intention of consumers in the 'older couple' life cycle stage.

Finally, our findings suggest that consumers' intention to make use of an online grocery service in the future are partly determined by adoption constructs, but that situational triggers, such as health problems or indeed the COVID-19 pandemic, seem to be needed to make consumers realize that they could actually benefit from using the service. For some consumers, it is only once they are confronted with the new situation—in which they actually order groceries via the Internet—that they (re)start thinking of the service in terms of the technology constructs. However, this finding is based on different samples of potential adopters and users at a single point in time. Further research could conduct a longitudinal analysis to determine how the evaluation of the variables changes over time.

**Supplementary Materials:** The following are available online at https://www.mdpi.com/article/10.3390/su13084144/s1, Table S1: Reliability for the full sample; Table S2: Convergent and divergent validity for the full sample; Table S3: Reliability for the full sample after removing FC4 and PR4; Table S4: Convergent and divergent validity for the full sample after removing FC4 and PR4; Table S5: Reliability for the potential adopters; Table S6: Convergent and divergent validity for the potential adopters; Table S7: Reliability for the potential adopters after removing PE4, EE4, FC4, PR4; Table S8: Convergent/divergent validity for the potential adopters after removing PE4, EE4, FC4, PR4; Table S9: Reliability for the users sample; Table S10: Convergent and divergent validity for the users sample; Table S11: Reliability for the users sample after the removal of FC4 and PR3; Table S12: Convergent and divergent validity for the users sample after the removal of FC4 and PR3.

**Author Contributions:** Conceptualization, E.V.D. and L.V.H.; formal analysis, E.V.D.; investigation, E.V.D.; methodology, E.V.D. and L.V.H.; supervision, L.V.H.; writing—original draft, E.V.D. and L.V.H.; writing—review and editing, L.V.H. All authors have read and agreed to the published version of the manuscript.

**Funding:** This research received no external funding.

**Institutional Review Board Statement:** Not applicable.

**Informed Consent Statement:** Not applicable.

**Data Availability Statement:** The data presented in this study are available on request from the corresponding author.

**Conflicts of Interest:** The authors declare no conflict of interest.

**Appendix A**

**Table 1.** Overview of the multi-item scales.

| Scale Items | References |
| --- | --- |
| PE1. I find online grocery services useful in my daily life.<br>PE2. Using electronic shopping of groceries saves much time.<br>PE3. Shopping groceries via the Internet is favorable as it makes me less dependent of opening hours.<br>PE4. There is a lot of money to save by buying groceries via the Internet [b]. | Hansen [48], Venkatesh et al. [23] |
| EE1. I find online grocery services easy to use.<br><br>EE2. It is hard to find the needed products when shopping groceries via the Internet.<br>EE3. With electronic shopping of groceries it is difficult to order products.<br>EE4. It is easy to check the availability of grocery items [b]. | Hansen [48], Hui and Wan [42], Venkatesh et al. [23] |
| SI1. Members of my family think that it is a good idea to buy groceries via the Internet.<br>SI2. Most of my friends and acquaintances think that shopping groceries via the Internet is a good idea. | Hansen [48,49], Venkatesh et al. [23] |
| FC1. I have the resources necessary to use an online grocery service.<br>FC2. I have the knowledge necessary to use an online grocery service.<br>FC3. Online grocery services are compatible with other technologies I use.<br>FC4. I can get help from others when I have difficulties using an online grocery service [a,b,c]. | Venkatesh et al. [23] |

**Table 1.** *Cont.*

| Scale Items | References |
|---|---|
| BI1. How likely is it that over the next year you will shop for groceries via the Internet? | Hansen [35,49], Hansen et al. [50] |
| HM1. Ordering groceries via the Internet is fun.<br>HM2. Ordering groceries via the Internet is enjoyable.<br>HM3. Ordering groceries via the Internet is very entertaining. | Venkatesh et al. [23] |
| PV1. Online grocery services are reasonably priced.<br>PV2. Online grocery services are good value for the money.<br>PV3. At the current price, online grocery services provide a good value. | Venkatesh et al. [23] |
| HB1. The use of an online grocery service has become a habit for me.<br>HB2. Ordering my groceries online has become natural to me. | Venkatesh et al. [23] |
| PR1. Return and exchange opportunities are not as good on the Internet as in the supermarket/non-Internet shop.<br>PR2. A risk when buying groceries via the Internet is receiving incorrect items.<br>PR3. I am concerned with the punctuality of the delivery time of online grocery shopping [c].<br>PR4. I am concerned with the quality of the products delivered when ordering from online grocery shopping [a,b]. | Hansen [35], Kurnia and Chien [41] |
| PTP1. I usually find myself pressed for time.<br><br>PTP2. I am often in a hurry.<br>PTP3. Usually there is so much to do that I wish I had more time.<br>PTP4. For different reasons, I do not have enough time for grocery retail shopping. | Van Kenhove and De Wulf [72];<br>Verhoef and Langerak [40] |
| I find shopping for groceries (in a physical store/supermarket) rather:<br>PSE1. Dull Exciting<br>PSE2. Not fun Fun<br>PSE3. Not amusing Amusing<br>PSE4. Not enjoyable Enjoyable | Spangeberg et al. [127] |
| SQ1. My online grocery store's employees are friendly and helpful.<br>SQ2. My online grocery store has fast check-out.<br>SQ3. My online grocery store provides adequate services (i.e., picking the orders, payment). | Hsu et al. [128], Boyer and Hult [90] |
| INN1. If I heard about a new information technology, I would look for ways to experiment with it.<br>INN2. Among my peers, I am usually the first to try out new information technologies.<br>INN3. In general, I am hesitant to try out new information technologies.<br>INN4. I like to experiment with new information technologies. | Agarwal and Prasad [94] |

[a]: These items were dropped for further analysis in the full sample. [b]: These items were dropped for further analysis in the potential adopters sample. [c]: These items were dropped for further analysis in the users sample.

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
