# Peer review of "Adoption and Usage of E-Grocery Shopping: A Context-Specific UTAUT2 Model"

_sustainability, doi:10.3390/su13084144_

Round 1

Reviewer 1 Report

This paper provides insight into the link between perceived risk, perceived time pressure, perceived in-store shopping enjoyment, innovativeness and online grocery shopping, in addition to the traditional UTAUT2 constructs. While interesting in light of the aim of the research, I found a couple of issues that need to be addressed before the paper is accepted for publication.

I’ve arranged my comments by the order in which they were motivated while reading the manuscript:

Introduction

(Line 35-36) It was indicated that “the uptake of online shopping proves to be highly product specific.” Please provide references to support this argument.

The authors claim that this study is the first to “validate the applicability of UTAUT2 in the online grocery shopping context”. However, I’ve found some paper with similar context:

Gert Human, Marius Ungerer, Jean-Axel J.C. Azémia (2020) Mauritian consumer intentions to adopt online grocery shopping: an extended decomposition of UTAUT2 with moderation, Management Dynamics: Journal of the Southern African Institute for Management Scientists, Vol. 29, No. 3.

Please check for the most updated references.

Theoretical background

Section 2.1 presented a review of the UTAUT2 model. I would suggest to remove this section due to the popularity of the UTAUTS model in empirical research and the length of the paper.

Discussion

One major insight in this article is a couple of constructs including SI, PR, PSE and EE is capable of discriminating between potential adopters and users (line 733-735). The reasoning behind is the just mentioned constructs are only significant in the full sample. There is a missing link between the finding of variables that are only significant in the full sample and their ability to discriminate between potential adopters and users. Please elaborate more on this discussion.

Conclusion

The conclusion is too long. Please summarize the main findings and pinpoint to the practical/policy implications.

Reviewer 2 Report

Congratulations for the paper and all the work behind it.

The topic is very interesting. The text flows very easily. The literature review is comprehensive. The hypotheses are clear and grounded in the literature. Despite the limitations that you mentioned in the paper regarding the data, the methodology is robust and adequate. The discussion is very well articulated and supported.

Having mentioned that, I have some comments/suggestions:

  1. You mention: “In order to determine how sustainable online grocery shopping is as a practice, it is crucial to have an in-depth understanding of its drivers” (lines 7-8); “This slow adoption is problematic, since online grocery shopping has the potential to address sustainability and other society-relevant concerns” (lines 43-44); “It is therefore of crucial importance to assess consumers’ hesitation to adopt online grocery shopping, and to further investigate the barriers as well as the drivers, so as to determine how sustainable e-grocery is as a practice” (lines 48-50). As far as the linkages with sustainability issues, this is everything you mention in the paper! The theme and linkages with your (central) topic are completely forgotten after the abstract and the introduction.
  2. You mention the need of triggers to make consumers to realize the benefits of online grocery service and start to use it. You also mentioned that the COVID-19 reinforces the importance of online grocery shopping. COVID-19 has been undoubtedly a huge trigger, forcing thousands/millions of people (around the world) to adopt and rely on online (grocery) service(s). Providing that you apply the survey in nov/dec 2014, perhaps with results not very different from applying it exactly in the same places in nov/dec 2019, I ask: how different would be comparing with nov/dec 2024? Do you think your managerial implications would be the same? In this particular case, in my humble opinion, the date of the survey, at least, can also be a limitation, with implication for the relevance of your findings and recommendations.
  3. It is nor clear to me the relevance of the “frequency of online ordering”, mentioned several times and part of the questionnaire, namely because there is no mention to the “frequency of onsite buying”. The same applies to the average amount spent.
  4. In this case, it seems questionable whether it is desirable to present the limitations within the conclusions (and within the text), and not in a separate section. The same for further research required.
  5. Finally, the size of the paper is not a concern, but consider to reduce/consolidate a few parts of the text.

Best regards

Round 2

Reviewer 2 Report

Dear authors, I am satisfied with your explanations and the improvements. Best regards